# Automated Service Height Fault Detection Using Computer Vision and Machine Learning for Badminton Matches

**DOI:** 10.3390/s23249759

**Published:** 2023-12-11

**Authors:** Guo Liang Goh, Guo Dong Goh, Jing Wen Pan, Phillis Soek Po Teng, Pui Wah Kong

**Affiliations:** 1School of Mechanical and Aerospace Engineering, Nanyang Technological University, Singapore 639798, Singapore; guoliang.goh@ntu.edu.sg; 2Physical Education and Sports Science Academic Group, National Institute of Education, Nanyang Technological University, 1 Nanyang Walk, Singapore 637616, Singapore; nie173748@e.ntu.edu.sg (J.W.P.); phillis.teng@nie.edu.sg (P.S.P.T.); 3Rehabilitation Research Institute of Singapore, Nanyang Technological University, 11 Mandalay Road, Singapore 308232, Singapore

**Keywords:** sports technology, robot umpire, computer vision, machine learning, system development

## Abstract

In badminton, accurate service height detection is critical for ensuring fairness. We developed an automated service fault detection system that employed computer vision and machine learning, specifically utilizing the YOLOv5 object detection model. Comprising two cameras and a workstation, our system identifies elements, such as shuttlecocks, rackets, players, and players’ shoes. We developed an algorithm that can pinpoint the shuttlecock hitting event to capture its height information. To assess the accuracy of the new system, we benchmarked the results against a high sample-rate motion capture system and conducted a comparative analysis with eight human judges that used a fixed height service tool in a backhand low service situation. Our findings revealed a substantial enhancement in accuracy compared with human judgement; the system outperformed human judges by 3.5 times, achieving a 58% accuracy rate for detecting service heights between 1.150 and 1.155 m, as opposed to a 16% accuracy rate for humans. The system we have developed offers a highly reliable solution, substantially enhancing the consistency and accuracy of service judgement calls in badminton matches and ensuring fairness in the sport. The system’s development signifies a meaningful step towards leveraging technology for precision and integrity in sports officiation.

## 1. Introduction

In badminton, players serve the shuttlecock to opponents across the court by striking it with a racket to start a rally. The Badminton World Federation (BWF) governs several types of service faults [1]. The most prevalent fault, known as “service fault: too high” (SFTH), occurs when the shuttlecock is hit higher than the permissible limit to gain an advantage. This fault is especially common in badminton doubles, where players aim to serve at the highest possible point, creating a flatter trajectory that is harder to return [2,3]. Historically, SFTH was identified by a service judge to determine whether the shuttlecock was struck above the server’s waist. This human judgement can cause problems such as non-standardized height limits favoring taller players and ambiguity in defining waist height [4]. In 2018, the BWF mandated that the service height must not exceed 1.150 m from the ground [5]. A fixed height service tool (FHST) aids judges in enforcement, but it does not provide evidence to back the service judge’s decisions and there is no publicly available data about its accuracy.

From a broader perspective, technologies have been employed to ensure fairness in sports [6]. Various technologies have been developed to make the judging process more accurate, efficient, and unbiased [7]. Additionally, the use of technology can alleviate the stress that referees or technical officials feel when trying to make the best judgement. One example is the video assistant referee (VAR) system that was used during the 2018 International Federation of Association Football (FIFA) World Cup. This system allowed for real-time communication with the referees on the field and was found to be more accurate than human decisions (98.3% for the system and 92.1% for humans, respectively) [8]. Other sports games such as the National Football League (NFL) and professional basketball also use video replay systems to review and confirm referee decisions. Recently, there have been attempts to use artificial intelligence in sports officiating [9]. In badminton, the instant review system (IRS) was first introduced in 2013 to aid in line call decisions, but it is not used for service fault detection. There was an attempt to use ultrasonic sensors to detect service height in badminton matches, but the system has several problems such as a limited scanning range and difficulty in detecting service events [10].

Machine learning techniques have been shown to be powerful tools in various fields including but not limited to manufacturing processes [11,12,13], robotics [14,15], unmanned vehicles [16,17], fault detection for quality inspection [18], and medical diagnosis [19]. Combining with computer vision, these techniques allow for the detection of objects of interest based on the images captured from camera systems using an advanced computer algorithm. Recently, computer vision and object detection techniques have also found applications in sports technology for match analysis [20,21,22,23], player’s performance analysis [24], and electronic umpiring [25,26]. In a recent study on video analysis of table tennis, the researchers have successfully demonstrated the real-time detection of the position of the tennis ball and players, and the types of events (bounces and net hits) [23]. In badminton, computer vision and machine learning have been used to achieve automated badminton action recognition for non-real-time post-match analysis [27,28,29]. Menon et al. proposed a machine learning framework for service fault detection; however, the challenge still remains as the framework runs at a relatively low sample rate (25.8 fps) [30]. To our best knowledge, there is currently no real-time vision-based technology with a high sample rate (>60 fps) that helps in the umpiring of service situations in badminton.

In the present study, we aimed to develop an automatic service height fault detection system using computer vision and object detection techniques to allow for real-time electronic service fault judging. In order to achieve a functional system that operates in real-time, the system requires an object detection model that is fast and able to identify the objects of interest such as players, rackets, shuttlecocks, and shoes accurately. Here, we adopted the You-Only-Look-Once v5 (YOLOv5) object detection model for the automated service height fault detection system [31]. An algorithm was developed to identify the hitting instance of the badminton service so that the height of the shuttlecock can be accurately retrieved. The accuracy of the calls made by humans and the proposed system were compared against the data acquired via a 3D motion capture camera system with high temporal resolution. This work contributes a novel approach to real-time electronic fault judging, promoting fairness and precision in the sport.

## 2. Experimental Methods

### 2.1. Development of Automatic Service Fault Detection System

#### 2.1.1. System’s Hardware Configuration

The proposed automatic service fault detection system consists of a two-camera system, a workstation, and an output display as shown in Figure 1a. One camera is used for each side of the court. The lens of the camera system is carefully selected so that the camera can be placed approximately 1.9 m away from the sideline of the court, near where the designated seat for the service judge is shown in Figure 1b. Figure 1c shows the viewing angle from behind the camera system. Also, the field of view of the camera is selected such that the camera can see the server and receiver in a service situation for the event categories which include singles, doubles, and mixed doubles matches. Figure 1d,e show the actual field of view of the left and right cameras in a badminton match. With the proposed camera system, the system can detect almost all types of service faults such as service fault too high, early server foot movement, undue hitting action, server stepping on the line, and early receiver movement. However, the focus of this study is mainly on the detection of SFTH.

#### 2.1.2. Working Principle and Camera Calibration

For an accurate determination of the shuttlecock height relative to the 1.150 m height limit, the camera has to be carefully calibrated to ensure that the center of the CCD sensor is placed at 1.150 m from the ground and the optical axis of the camera is always pointing along the 1.150 m plane from the ground, as shown in Figure 2. Figure 3 shows the possible service fault detection error due to poor camera placement and alignment. These examples show the possible detection errors where an illegal service (SFTH) is deemed legal by the camera system when the camera is misaligned. The opposite can also happen where a legal service is deemed too high by the camera system if the misalignment of the camera occurs in the reversed direction. In order to properly calibrate the camera, a camera gimbal with two angular degrees of freedom (roll and pitch) has been developed. The camera height is first manually adjusted to 1.150 m from the ground using a 1.150 m long reference stick. The cameras are then calibrated for roll and pitch angles using two reference sticks that are placed in the field of view of the cameras.

#### 2.1.3. Object Detection Model and Generation of Datasets for Model Training

An object detection model is required to locate the position of the players and shuttlecock in real-time. Therefore, it is important that the object detection model used in this work is fast enough for the system to work with high temporal resolution. As such, the YOLOv5 object detection model is used due to the high inference speed and high accuracy [31,32]. Using this object detection model, we can obtain information about the position and displacement of the shuttlecock, which can then be used to infer the hitting instant of a badminton service.

To facilitate good detection for the object detection model, a high-quality image dataset is required. The camera was placed at the side of the court, where the service judge is seated at a height of around 1.150 m. A total of six venues were used for the data acquisition to ensure a good diversity of hall environments for a better generalization of data. Collectively, 19 badminton players, including 14 university team players, were involved in the data collection for object detection model training. Among them, 16 were male players. A summary of the dataset was shown in Table 1.

To ensure a good mix of different service styles, the service conditions of the men’s singles, men’s doubles, and women’s doubles were recorded. This ensured that the common service conditions (Figure 4), which comprised backhand low serves, backhand flick serves, forehand low serves, and forehand high serves, were included in the dataset.

Capturing footage of player serving has a downside, which is that the label counts for the shuttlecock are much lower compared to the other labels. The imbalance dataset is not ideal as it would cause the object detection model to have less of a chance to learn to detect the shuttlecocks, causing poor accuracy at recognizing the shuttlecock. Since the key feature of the system is to detect whether the shuttlecock is struck above the 1.150 m height limit, the ability of the objection detection model to recognize the shuttlecock is of utmost importance. To balance the dataset, 1900 images that contain only shuttlecocks were taken to increase the number of instances of the shuttlecocks (Figure 5). Overall, the training and validation dataset has a class distribution, as shown in Figure 6. 

#### 2.1.4. Service Fault Detection Algorithm

The service height is determined by the maximum height of the shuttlecock when it is being hit by the racket during a badminton service. The hitting instant refers to the moment when the racket first comes into contact with the shuttlecock. Thus, the determination of the hitting instant is of utmost importance. In this work, the hitting instant (N^th^ frame) is determined by the large horizontal displacement of the shuttlecock as compared to the next frame (N+1^th^ frame). Since the rule of badminton mandates the whole of a shuttlecock to be below the 1.15 m height limit upon being hit, the highest point of the shuttlecock should then be used to evaluate whether the shuttlecock has exceeded the height limit. The highest point of the shuttlecock can be evaluated based on the N^th^ frame, as depicted in Figure 7. 

An overview of the service fault detection system algorithm is depicted in Figure 8. Essentially, the acquired image from the camera is first resized to a dimension of 640 pixels × 640 pixels before feeding into the object detection model. This is vital because running inferences with the object detection model at the native image dimensions can severely slow down the inference speed. The output from the object detection model such as the label and the bounding box of the shuttlecock will be obtained and used to determine the displacement of the shuttlecock. The hitting event is then determined when a large horizontal displacement is detected. The height information about the shuttlecock at the hitting instant can then be evaluated based on the bounding box of the shuttlecock at the N^th^ frame, as depicted in Figure 7.

The algorithm for detecting SFTH in badminton matches involves a series of steps:

Image acquisition and preprocessing: The algorithm starts by acquiring the latest image, termed as the N+1^th^ frame, and then performs necessary preprocessing image inferencing.Object detection: In this step, the trained YOLOv5 object detection model is deployed to identify the shuttlecock within the image.Shuttlecock centroid calculation: If the shuttlecock is detected, the algorithm calculates the centroid of the shuttlecock for accurate positioning.Centroid comparison: The algorithm then compares the current shuttlecock centroid (in the N+1^th^ frame) with the centroid from the previous frame (N^th^ frame) to track movement.Displacement analysis: In case a significant displacement of the shuttlecock is observed, the bounding box of the shuttlecock from the previous frame (N^th^ frame) is analyzed to determine its maximum height during the service (h_max_).Height verification: This step involves checking whether the maximum height (h_max_) of the shuttlecock exceeds the official service height limit of 1.150 m.Fault decision: Based on the height analysis, if h_max_ is greater than 1.150 m, a service fault is declared; otherwise, the service is deemed legal.

#### 2.1.5. Workstation Configuration

The service fault detection algorithm was coded in Python 3.6 and executed using a workstation. The workstation consists of an AMD Ryzen 7 3700X CPU, an image acquisition card that can handle two high-speed cameras, two units of Gigabyte RTX 3080 Eagle OC GDDR6, and 32 GB of DDR4 RAM. The operating system of the workstation is Ubuntu 18.04. In order to speed up the image processing and inference time, Nvidia DALI and TensorRT algorithms were used for resizing the images and the image inferencing process, respectively. In this work, the system was set to run at 70 fps.

### 2.2. Comparative Analysis Using Motion Capture System as Benchmark

The performance of the service judge and the developed system were benchmarked against the data collected using a high-speed motion capture system for comparison. A Vicon 3D motion capture system (United Kingdom) was used in this work (Figure 9a). The motion capture system consists of 8 cameras that acquire data at a sampling rate of 200 Hz. Both the shuttlecock and the racket were equipped with three retro-reflective markers each (Figure 9b,c). For this comparative analysis, the system is operated with only a single camera. A digital model with the same dimensions as the actual test shuttlecock was used for its digital reconstruction. The highest point of the shuttlecock was then calculated via the reconstruction of a 3D model based on the acquired coordinates of the markers (Figure 9d). The system and the FHST were placed 5 m away from a fixed location of the server to simulate actual tournament settings. 

The service height was determined by analyzing the shuttlecock horizontal speed while employing a similar concept to that described in Figure 7. Figure 10 shows the shuttlecock dynamics during a backhand short service, with the datapoints acquired at a period of 5 ms (200 Hz). The hitting instant (N^th^ datapoint) was determined when a sudden jump in the shuttlecock speed (equivalent to large displacement) was observed. The height of the shuttlecock at the N^th^ datapoint was determined by the reconstruction of a 3D model based on the acquired coordinates of the markers (Figure 9d).

This study was approved by the Nanyang Technological University Institutional Review Board (IRB-2022-870). A total of eight participants were involved and provided written informed consent to participate in the comparative analysis using the 3D motion capture system. Three of them were professional service judges (SJ 1–3) whereas the rest (Non-SJ 1–5) did not have any experience using the fixed height service tool. A short training was conducted for the inexperienced participants. A laser leveling system was used to calibrate the fixed height service tool and the developed system. Each participant made judgement calls for 20–40 services. We engaged two badminton players to be the servers for the tests, and they took turns to perform the services for each round. The servers would attempt to serve close to the service height limit with the aid of the laser leveling device. Figure 9e shows the service height distribution of the serves in this study. In this study, only backhand low service was evaluated since it is the most common type of service used by professional players in tournaments. A summary of the tests and participants’ information can be found in Table 2.

## 3. Results and Discussions

### 3.1. Object Detection Model Training for Service Fault Detection

The results obtained from the training of the YOLOv5 object detection model is summarized in Figure 11. The steady decrease without bouncing back up at the later stage indicates that overfitting has not occurred. The lower validation loss (Figure 11b) compared to the training loss (Figure 11a) was due to the use of data augmentation. The model, which was trained for 580 epochs, achieved a mean average precision of mAP@0.5 of 0.99 (Figure 11c). This means that on average, among all the classes, the percentage of correctly detecting the object and locating the bounding box overlapping more than half the area of the object was 99%. The precision-recall curve shows the trade-off between precision and recall for different thresholds (Figure 11d). A large area under the curve represents both high recall and high precision, where high precision relates to a low false positive rate, and high recall relates to a low false negative rate. The high scores for both show that the classifier is returning accurate results (high precision), as well as returning a majority of all positive results (high recall). Figure 11e shows an example of the inferenced output produced by the trained YOLOv5 model. It was observed that the model is able to infer objects such as shuttlecocks, players, shoes, and rackets correctly. The ability to achieve high recall and precision also means that the system will produce less missed and false service detections. The trained model was cross-validated using another dataset with 968 untrained images. It was observed that the trained model was still able to achieve a mean average precision mAP@0.5 of 0.956 (Appendix A), indicating a high level of accuracy and robustness in detecting service faults under varied conditions. This performance demonstrates the model’s strong generalization capabilities, affirming its effectiveness in real-world applications beyond the initial training set. The utilization of the object detection model for the detection of service faults in real-time is demonstrated in Appendix A. Appendix A are demonstrations showing that the system has detected a service that was deemed legal and too high, respectively. 

### 3.2. Comparative Analysis Using Motion Capture System as Benchmark

In order to better compare the system’s performance with that of the human judges, a comparative study was conducted using the motion capture system. In this work, we evaluated the accuracy of the system based on detected services. We assessed 204 services for the system and 255 services for humans, with an approximately equal division: close to half of these services were categorized as legal, and the remaining half were categorized as faulty. The number of services assessed for the system were slightly lower due to some missed detections, which were caused by several reasons to be discussed in the Limitations Section.

Here, we are introducing two performance metrics known as the “range of confusion (RoC)” and “deviation of midpoint of confusion (MoC)” (Figure 12). These metrics are utilized to assess the service calls made by both humans and the system. The RoC is characterized as the vertical range within which a service has been identified as both legal and faulty at different instances. The lower bound of the RoC is determined from the lowest height from the human/system called the fault dataset or the service height limit, whichever is lower. In contrast, the upper bound of the RoC is determined from the service height limit or the highest height from the human/system called the legal dataset, whichever is higher. The midpoint of confusion (MoC) is defined as the median height of the data that fall within the range of confusion. The RoC measures the consistency level of the judgement calls, whereas MoC is used as a reference to give an indication of the deviation of the range of confusion to the 1.150 m service height limit. Essentially, a larger RoC indicates a poorer consistency in the judgement call, and a larger deviation of MoC potentially indicates larger parallax or system errors.

A boxplot is used to visualize the distribution of the calls for each round (Figure 13a,b). For a perfect judgement, the boxplot of the legal call should be below the 1.150 m service height limit whereas the boxplot of the fault call should be above the 1.150 m service height limit. In other words, an arbitrary line at the 1.150 m height limit should split the two boxplots in an ideal situation. 

To evaluate the consistency of the service calls, the RoC for each round is plotted for comparison (Figure 13c,d). Human calls vary from human to human. Some participants (non-SJ 1, 3, and 5) were also more consistent with their calls, as observed from the smaller RoCs (23.6 mm, 36.4 mm, and 21.8 mm, respectively), while others have RoCs greater than 50.0 mm (Figure 13c). It was also noted that the RoCs are extremely large for some of the participants (non-SJ 4 and SJ 1). It is also observed that the MoCs vary drastically from person to person. Some of the MoCs fall at approximately 1.170 m (non-SJ 1,4,5), and some fall at about 1.120 m (non-SJ 2). The large differences could be partly attributed to parallax error, where each participant has its own way of using the fixed height service tool. For non-SJ 1,4, and 5, another possible reason is that they were more conservative in making service fault calls, thus resulting in a higher MoC. 

In comparison with human judges, no overlap between the boxplots of the legal and fault call is observed for the calls made by the system (Figure 13b). The RoC of the system for each round is generally smaller when compared to the human call (Figure 13d), which suggests that the system can reliably and consistently make better calls compared to humans. 

Figure 14a,b show the distribution of the human and system’s service judgements, respectively. The system typically makes more fault calls than legal calls above the service height limit. In contrast, the legal call count is higher than the fault call count by up to 1.175 m for humans, suggesting that humans tend to be more conservative in making a fault call in the 1.150–1.175 m range. Humans also falsely identified services below 1.150 m as service faults more often compared to the system.

The services were grouped into various height ranges to understand the performance of the human judges and system at each range, as depicted in Figure 14c. A lower accuracy suggests that there are discrepancies in the classification of services, either in terms of false positives (misclassifying legal services as faults) or false negatives (failing to identify fault services). When the accuracy is lower in the height range below 1.150 m, it indicates that more service calls which should be categorized as legal services are instead being mistakenly identified as faults. In contrast, when the accuracy is lower in the height range above 1.150 m, it indicates that more service calls which should be categorized as fault services are not being correctly identified as faults. It was found that the system has a higher accuracy at every height range. For the service executed below 1.150 m, the performance between the human and the system is comparable, with the system having a slight edge over the human. However, for the services within the range of 1.150–1.750 m, the system outperformed humans by 1–3.5 times. For instance, the accuracy of the human between 1.150 and 1.155 m is merely 16%, whereas the accuracy of the system is 58% for the same range. Humans achieved an accuracy of 80% above 1.175 m, whereas the system was able to achieve an accuracy of 100% above 1.170 m. The higher service fault detection accuracy of the system also implies that more service fault calls would be expected if it were to be implemented in an actual tournament.

The detection accuracy of the system depends on the spatial and temporal accuracy of the detection. The spatial accuracy is determined by factors such as camera calibration and the accuracy of the bounding box detection (Figure 15a). In contrast, the temporal accuracy is determined by the camera and system settings, such as the threshold speed and processing speed, which affect the ability to precisely determine the hitting event. It should also be highlighted that the shuttlecock dynamic during a service also plays a part in determining the detection accuracy of the system. Figure 15b shows the shuttlecock dynamics during a typical backhand short service and it can be used to explain how various factors such as the threshold speed and temporal resolution of the system affect the height detection error, and hence the inaccuracy. The relative height of the shuttlecock shown in the plot is calculated with respect to the actual hitting instant, meaning to say that the relative height at the actual hitting instant is 0 mm, which can be observed as the orange line intersects the x-axis (time axis). The detection of a hitting event (when the shuttlecock touches/leaves the racket) is determined by checking if the shuttlecock speed exceeds a predefined threshold speed on the (N+1)^th^ frame. The height information of the shuttlecock captured by our system is then determined from the N^th^ frame. In Figure 15b, where the shuttlecock has a vertical motion (change in height) prior to being hit, an error is observed. The magnitude of the error will depend on temporal accuracy, which is when the hitting instant is being detected. Figure 15b-i and 15b-ii are two different cases that show how the temporal resolution and the misalignment of the hitting event with the frames affect the magnitude of the error. The temporal accuracy of the hitting instant will depend on the temporal resolution (how high the fps is) and the threshold speed. The higher the temporal resolution (higher fps), the smaller the time gap between frames; hence, better temporal accuracy can lead to smaller error. Also, it is important to note that the speed of the vertical motion prior to a service also influences the detection error proportionally. Usually, the faster vertical motion of the shuttlecock during a service leads to a higher detection error. Thus, it is vital to improve the system processing speed so that the temporal accuracy of the hitting event can be improved. Increasing the system processing speed not only improves the temporal accuracy, but also allows for a model with a larger input (higher image resolution), resulting in an improvement in the spatial accuracy of the detection. 

## 4. Limitations

It should be noted that the system accuracy presented in this work may not reflect the performance of the system at the actual competition venues. In the tournament settings, the system may exhibit better accuracy. There are three reasons why the system may show a lower accuracy in the lab condition compared to the actual settings. Firstly, the object detection model did not give good detections in the lab because the model has not been trained with images acquired from the lab environment. Secondly, the shuttlecock was attached with three retro-reflective ball markers that could affect the detection using the object detection model. Thirdly, the lighting in the lab environment was not as bright (500 lux) as that of the tournament location’s environment (1500 lux). A slower shutter speed was used in the lab environment to compensate for the low brightness, thus causing the moving shuttlecock to appear less sharp in the acquired images as compared to the tournament environment. The blurry shuttlecock can result in poorer shuttlecock detection by the object detection model, thus lowering the accuracy of the system in the lab environment. Nevertheless, the system still outperformed human judges in this suboptimal test environment.

Nonetheless, there are some limitations in the current two-camera setup when deployed in actual tournaments. For instance, the shuttlecock may be blocked by the server’s hand (Figure 15a-ii) or female players in the case of mixed doubles. A four-camera setup may potentially reduce the blind spot and overcome the problem. Additionally, the object detection model might sometimes yield false positives, mistakenly identifying white objects of similar sizes as shuttlecocks. To address this limitation, we have integrated a filtering algorithm designed to eliminate such false detections and improve overall accuracy. Also, accurately distinguishing players from non-player individuals is crucial for practical application. While a filtering algorithm to identify players has been developed, it is not discussed in this manuscript, as it falls outside the scope of this manuscript. More optimizations would be needed to speed up the code to lower the hardware requirement. Also, the proposed method using monocular vision does not directly provide precise information of the height of the shuttlecock. However, our initial unpublished research indicates that with appropriate calibration, an estimation model can accurately predict the shuttlecock’s height.

In our comparative analysis between human judgment and the system, benchmarked against the gold standard of a high sample-rate motion capture system, we did not evaluate the impact of distance on the accuracy of service judgment calls by both the human and the system. Also, our analysis was limited to the accuracy of service judgment calls for backhand low serves. The humans’ and system’s accuracies might differ from the findings in the current study for other service types, such as forehand high serves, due to the distinct shuttlecock kinematics associated with each serve.

## 5. Conclusions

The advent of artificial intelligence has enabled the development of an automated umpiring system. Here, we report the development of the automated service fault detection system using computer vision, the YOLOv5 object detection model, and an in-house developed algorithm to detect the hitting instant of the service. The object detection model was trained with high quality data including those obtained from actual competition environments. This has resulted in a mAP@0.5 of 0.99, suggesting that the model is capable of detecting shuttlecocks, players, rackets, and shoes with good accuracy. The system outperformed human judges in the comparative study, demonstrating its potential in improving the quality of the service judgement in an actual competition setting. As the system is able to eliminate human bias, it can give a more consistent call with higher accuracy. Future work would include exploring the pose estimation model and optimizing the algorithm to increase the system processing speed. It is also important to investigate the system’s performance on the detection of faults due to server undue action, the server’s and receiver’s feet movement, and players’ feet stepping on the line.

## 6. Patents

A patent application has been submitted for this work. Patent application number: PI2023005746. Application type: Patent. Application title: System for detecting one or more faults of a server and/or receiver during a service in a badminton game and method therefor. Submission date: 22 September 2023.

## Figures and Tables

**Figure 1 sensors-23-09759-f001:**
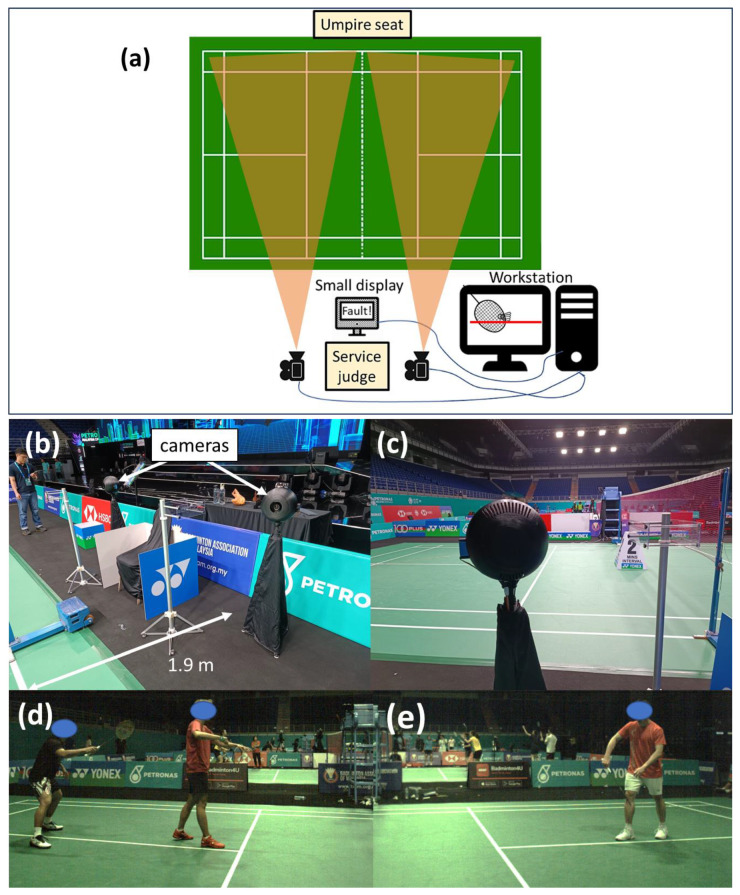
Schematic showing (**a**) proposed automatic service fault detection system and the approximated field of views of the cameras, (**b**) the actual setup of camera system captured in the Malaysia Open 2023, (**c**) viewing angle from behind the camera, (**d**,**e**) actual viewing perspectives of the left and right cameras, respectively.

**Figure 2 sensors-23-09759-f002:**
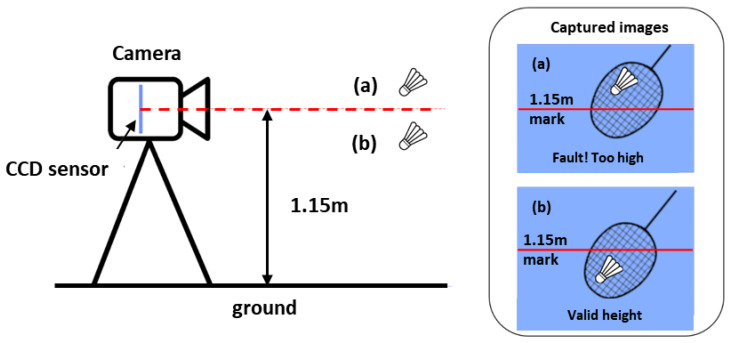
Schematic showing the working principle of service fault detection using the proposed camera configuration. Examples of (**a**) service fault too high (SFTH) and (**b**) legal service.

**Figure 3 sensors-23-09759-f003:**
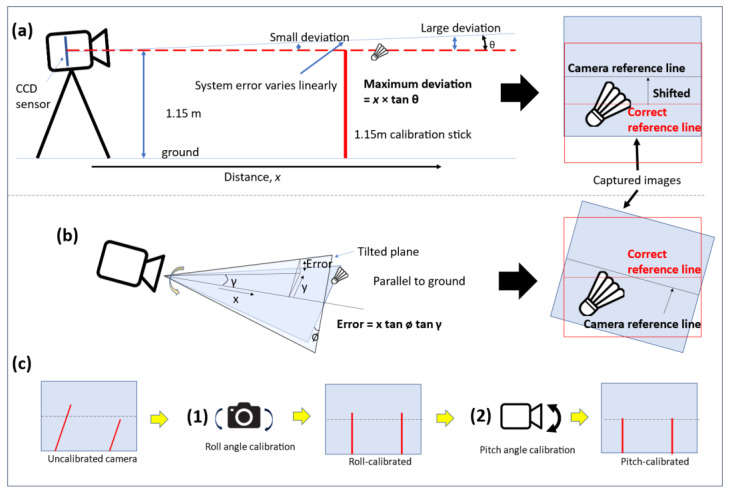
Schematic showing the detection errors due to (**a**) pitch and (**b**) roll angles’ misalignment. (**c**) Calibration steps for the camera system (white lines and red lines represent center of CCD sensor and the calibration sticks, respectively).

**Figure 4 sensors-23-09759-f004:**
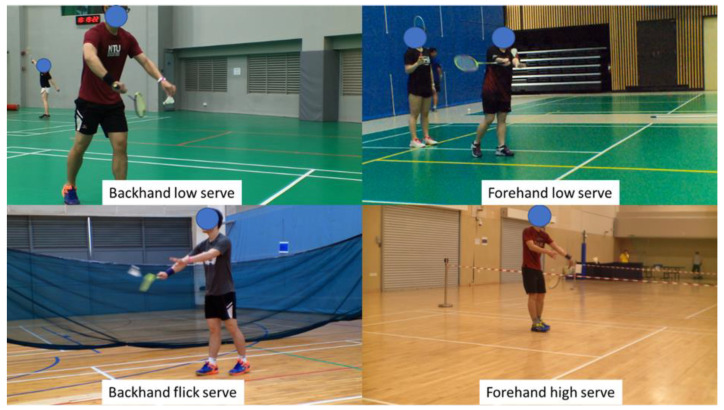
Various types of badminton service captured at different venues for the datasets.

**Figure 5 sensors-23-09759-f005:**
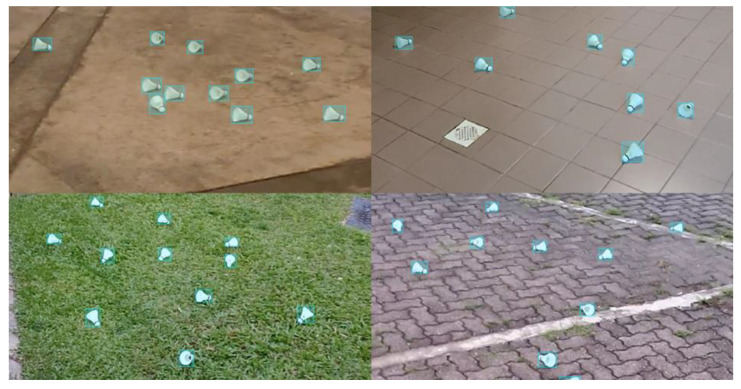
Shuttlecock-only images were added to increase the label counts of shuttlecocks to balance the dataset.

**Figure 6 sensors-23-09759-f006:**
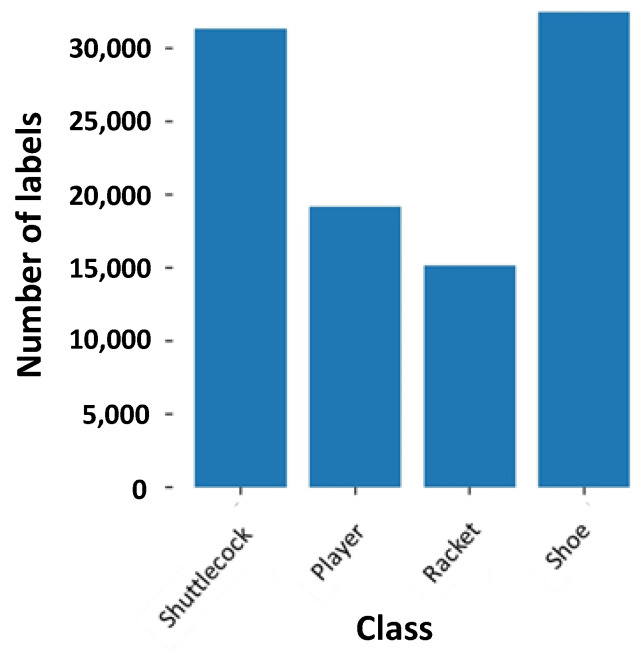
Distribution of the classes in the dataset.

**Figure 7 sensors-23-09759-f007:**
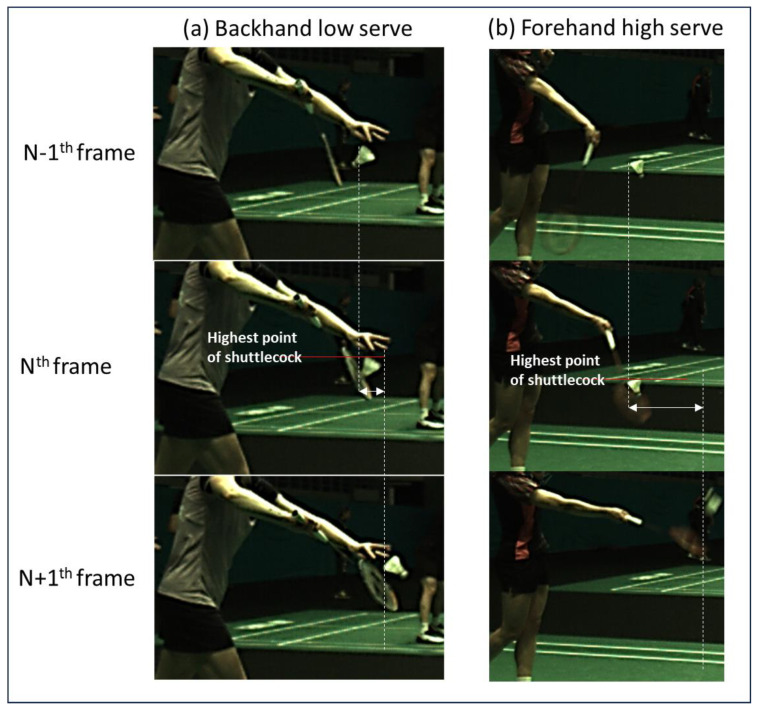
Image sequences (acquired at 70 fps) showing the hitting instants in (**a**) backhand low serve and (**b**) forehand high serve situations and the determination of the shuttlecock’s maximum height at the hitting instant (N^th^ frame). The hitting instant (N^th^ frame) is determined by the large horizontal displacement of the shuttlecock as compared to the next frame (N+1^th^ frame).

**Figure 8 sensors-23-09759-f008:**
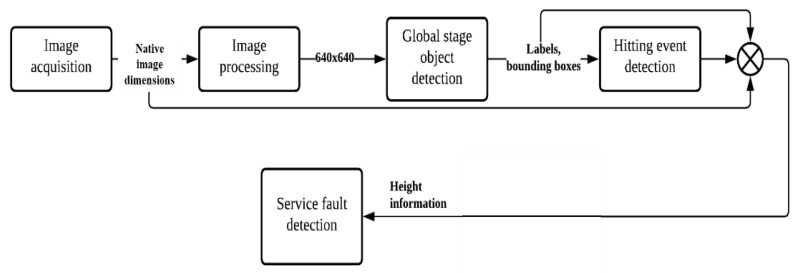
Overview of service fault detection algorithm.

**Figure 9 sensors-23-09759-f009:**
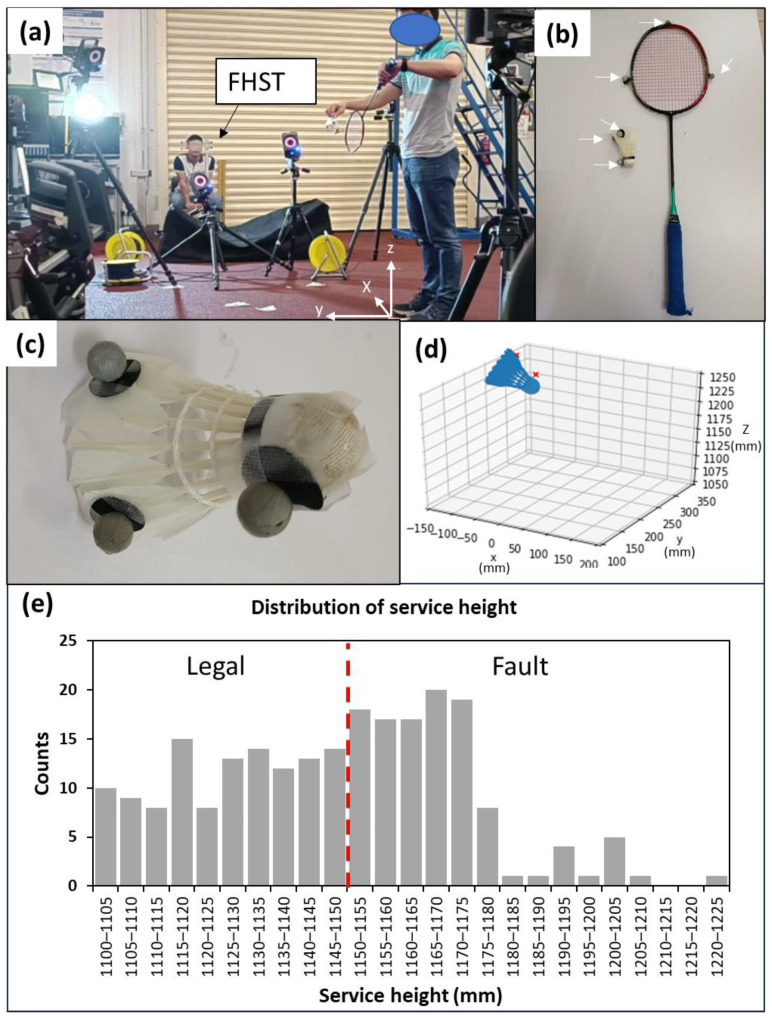
Images showing (**a**) the experimental setup of the motion capture system and the fixed height service tool (FHST), (**b**,**c**) white arrows pointing at the reflective markers attached to the racket and the shuttlecocks, (**d**) the reconstruction of 3D model using coordinates of markers (red crosses), and (**e**) service height distribution. The red dashed line separates the legal and fault services.

**Figure 10 sensors-23-09759-f010:**
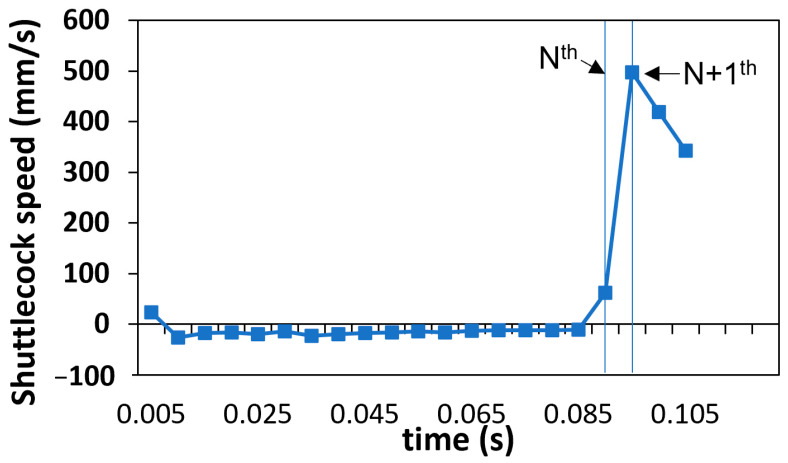
Graph showing the shuttlecock’s horizontal speed during a backhand short service.

**Figure 11 sensors-23-09759-f011:**
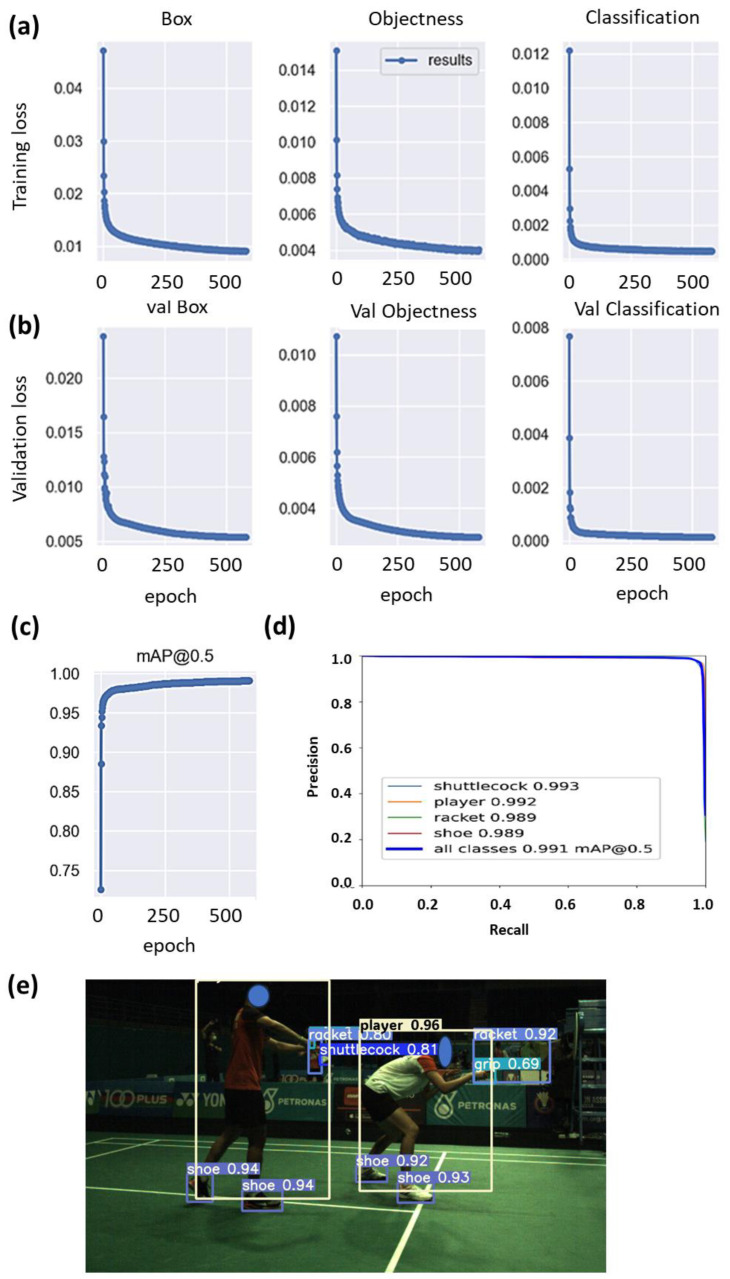
Graphs showing (**a**) the training losses and (**b**) validation losses for box, objectness, and classification. (**c**) Graph showing mAP at threshold = 0.5, reaching 99% after 580 epochs, (**d**) precision-recall curve of the trained model. Large area under the curve indicates high precision and high recall, (**e**) image showing the inference output of the trained model.

**Figure 12 sensors-23-09759-f012:**
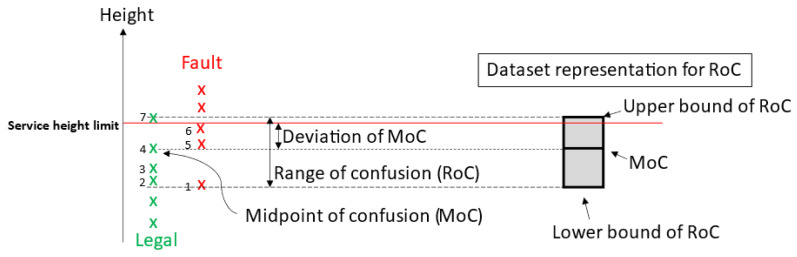
Diagram illustrating the definitions of the range of confusion, the deviation of midpoint of confusion, and the dataset representation for range of confusion (RoC).

**Figure 13 sensors-23-09759-f013:**
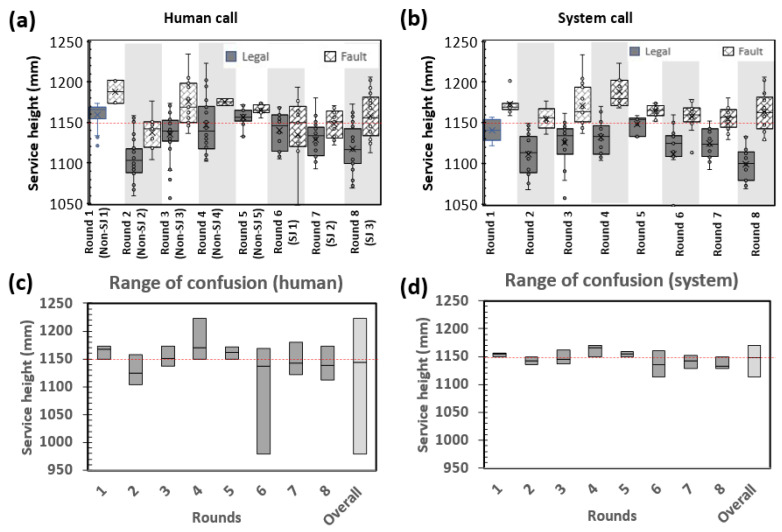
Boxplots of legal and fault calls made by (**a**) human and (**b**) system. Chart showing the region of confusion for (**c**) human and (**d**) system.

**Figure 14 sensors-23-09759-f014:**
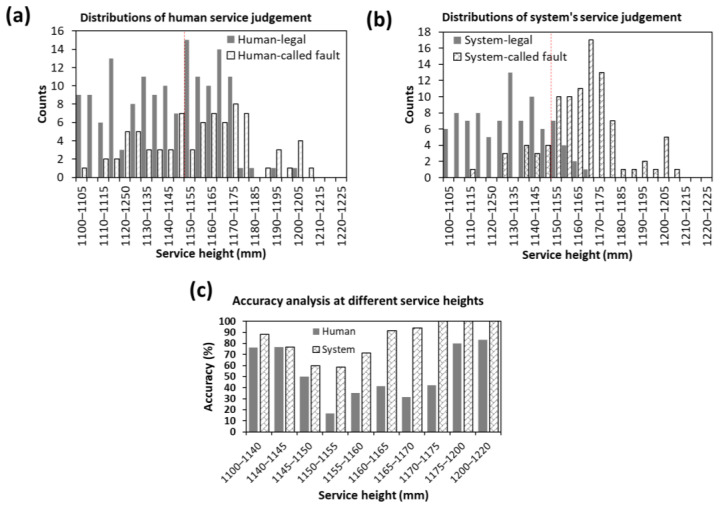
Histograms showing distribution of legal and fault calls made by (**a**) humans and (**b**) system. (**c**) Bar chart showing the accuracy of human and system calls at different service heights.

**Figure 15 sensors-23-09759-f015:**
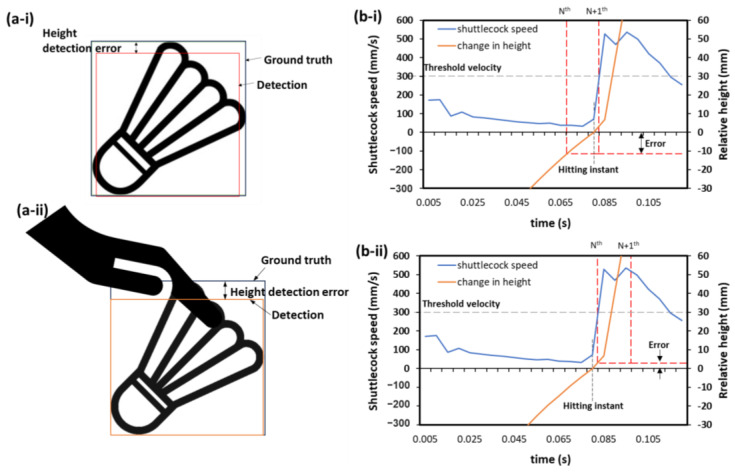
(**a**) Schematic showing the possible bounding box detection error due to (**a-i**) the inaccuracy of the object detection model’s prediction and (**a-ii**) occlusion, and (**b**) graphs illustrating the effect of processing speed and service style leading to height detection error. The relative height refers to the vertical distance between the measured shuttlecock height at N^th^ frame relative to the actual shuttlecock height at the hitting instant. (Note: the threshold velocity in the graphs is for illustration purposes only, the exact threshold is not disclosed.)

**Table 1 sensors-23-09759-t001:** Information about dataset for object detection model training.

Details of Dataset	Values
Image resolution	1152 pixels × 648 pixels
Number of venues	6
Number of players	19
Total number of images	25,235 images
Number of images in training set	18,588
Number of images in test set	4647
Ratio of train: test sets	80:20

**Table 2 sensors-23-09759-t002:** A summary of participant information.

Rounds	Gender	Experience Level	Nomenclature
1	Male	Inexperience	Non-SJ 1
2	Male	Inexperience	Non-SJ 2
3	Male	Inexperience	Non-SJ 3
4	Male	Inexperience	Non-SJ 4
5	Male	Inexperience	Non-SJ 5
6	Female	Professional	SJ 1
7	Male	Professional	SJ 2
8	Male	Professional	SJ 3

## Data Availability

The data presented in this study are openly available in the NIE Data Repository [https://doi.org/10.25340/R4/A2AE4H].

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
