# Peer review of "Automated Service Height Fault Detection Using Computer Vision and Machine Learning for Badminton Matches"

_sensors, 2023, doi:10.3390/s23249759_

Round 1

Reviewer 1 Report

Comments and Suggestions for Authors

The article covers an interesting topic that is relevant to Sensors. In my opinion, some points should be clarified before final acceptance.

1. The authors generated a dataset for YOLOv5 model training in the experiment. What is the number of training and test sets?

2. How to avoid detecting non-player humans in practical applications? The author should give some discussion.

3. Language needs improvement. 

4. Cross-validation of the data sets is needed to extensively evaluate the robustness of the proposed method for detecting shuttlecocks, players, rackets, and shoes in images.

Comments on the Quality of English Language

Language needs improvement. 

Reviewer 2 Report

Comments and Suggestions for Authors

This paper has proposed an efficient and accurate automated service height fault detection system based on computer vision and machine learning techniques for badminton matches, which shows better accuracy as compared to the human judges’ result. The experimental results are convincing and show its good performance on the target goal. More specific comments on this paper are summarized as below:

(1)   The proposed system combines the computer vision methods and machine learning method (using Yolov5 model for object detection), achieving excellent accuracy results compared to experienced human judges.

(2)   The dataset collection and labeling used for Yolov5 model training is well prepared, by inviting many badminton players to record the data, doing the dataset labeling and then used for model training. The model training accuracy can achieve 99% mAP, which is great work for this application.

(3)   The proposed system has been tested in a real badminton game and the tested result is promising and persuasive.

(4)   The authors also provide lots of experimental data analysis to show why the performance of the proposed system is better than that of human judges, which is a good reference for researchers who are interested in this topic.

Reviewer 3 Report

Comments and Suggestions for Authors

This paper proposes developed an automated fault detection system that employed computer vision and machine learning specifically utilizing the YOLOv5 object detection model.

The proposed method can pinpoint the shuttlecock hitting event to capture its height information. To assess the accuracy of the new system, the proposed method benchmarked the results against a high sample-rate motion capture system and conducted a comparative analysis with 8 human judges that used a fixed height service tool in a backhand low service situation. Overall, this paper is well-structured and presents some promising results. Here are some suggestions for further improving the quality of this paper:

(1) The practical value of this research work should be clarified and highlighted in the Abstract, which can help readers understand the engineering background of this research work.

(2) The resolutions of the figures should be improved. For example, Figure 1 (a), Figure 2, Figure 4 and Figure 10 are of low quality. In addition, the figure annotations should have a unified format.

(3) Please provide a detailed and intuitive explanation of how the proposed algorithm is applied to the fault detection.

(4) The authors did a good literature review on the current research progress. It is suggested to summarize the research gaps before presenting and introducing this paper's main contributions and novelty. In addition, it is suggested to include some discussions on the emerging areas of Kalman filter and artificial intelligence for fault diagnosis in various industrial applications, especially in terms of Feature Extraction of Multi-sensors for Early Bearing Fault Diagnosis Using Deep Learning Based on Minimum Unscented Kalman Filter.

(5) It is suggested to include some recommendations for future work at the end of the Conclusion.

Comments on the Quality of English Language

Minor editing of English language required
